# Nuclear Fusion Diamond Polishing Dataset

**Antonios Alexos**[1]    **Junze Liu**[1]    **Shashank Galla**[2]    **Sean Hayes**[3]
**Kshitij Bhardwaj**[3]    **Alexander Schwartz**[3]    **Monika Biener**[3]
**Pierre Baldi**[1]    **Satish Bukkapatnam**[2]    **Suhas Bhandarkar**[3]
[1]University of California, Irvine
[2]Texas A&M University
[3]Lawrence Livermore National Lab

## Abstract

In the Inertial Confinement Fusion (ICF) process, roughly a 2mm spherical shell made of high-density carbon is used as a target for laser beams, which compress and heat it to energy levels needed for high fusion yield in nuclear fusion. These shells are polished meticulously to meet the standards for a fusion shot. However, the polishing of these shells involves multiple stages, with each stage taking several hours. To make sure that the polishing process is advancing in the right direction, we are able to measure the shell surface roughness. This measurement, however, is very labor-intensive, time-consuming, and requires a human operator. To help improve the polishing process we have released the first dataset to the public that consists of raw vibration signals with the corresponding polishing surface roughness changes. We show that this dataset can be used with a variety of neural network based methods for prediction of the change of polishing surface roughness, hence eliminating the need for the time-consuming manual process. This is the first dataset of its kind to be released in public and its use will allow the operator to make any necessary changes to the ICF polishing process for optimal results. This dataset contains the raw vibration data of multiple polishing runs with their extracted statistical features and the corresponding surface roughness values. Additionally, to generalize the prediction models to different polishing conditions, we also apply domain adaptation techniques to improve prediction accuracy for conditions unseen by the trained model. The dataset is available in
`https://junzeliu.github.io/Diamond-Polishing-Dataset/`.

## 1   Introduction

In the wake of significant breakthroughs in achieving ignition as highlighted by [Abu-Shawareb et al., 2024, Moses, 2010], the Inertial Confinement Fusion (ICF) program at the National Ignition Facility (NIF) has shifted its focus towards establishing a viable, high yield fusion platform. The series of successful experiments, achieving gains greater than one subsequent to the initial demonstration of ignition, signal the onset of a robust research phase. This advancement underscores the importance of enhancing the energy output, which not only facilitates exploration within the realms of high energy density physics previously beyond reach but also lays down the foundational principles for conceptualizing high gain (significantly greater than 10) strategies essential for the efficacious harness of energy through nuclear fusion.

Achieving an optimized laser energy output from the apparatus at the NIF is crucial for the amelioration of net energy yield from the experiments conducted therein. This necessitates continuous improvements in both the design and manufacturing technology of the targets – the experimental entities subjected to laser irradiation for fusion. The construction quality of these ICF targets, which involves complex assemblies of numerous precision-engineered components, is imperative for ensur-

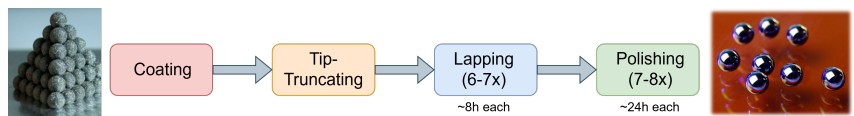

Figure 1: Overview of the ICF shell fabrication process

ing the maximized and efficient use of the deposited laser energy. These assemblies are crucial as they ensure the formation of a defect-free solid phase of deuterium-tritium (DT) fuel when prepared at approximately 19K, an aspect extensively discussed by Hamza [2005].

For high-yield ICF experiments, it is imperative to generate and maintain high pressures and temperatures, lasting on the order of several nanoseconds, to overcome the Coulomb barrier inhibiting the fusion of DT nuclei. This condition is facilitated by substantial compression of a hollow sphere of solid fuel, leading to the ignition of a burn wave that propagates throughout the fuel mass, a concept detailed in the works of [Lindl et al., 1992, Lindl, 1995, Hurricane et al., 2023]. This compression is primarily achieved through the enormous reactionary implosive force generated by the extremely rapid ablation of a capsule, colloquially termed as the 'rocket,' housing the DT fuel. Made predominantly of low atomic number (low Z) materials such as diamond-like high-density carbon (HDC) – as detailed in the works of [Haan et al., 2011, Ross et al., 2015, Clark et al., 2018] – this capsule undergoes instantaneous compression to a fraction of its original size at the culmination of the implosion sequence.

Given the capsule's central role in achieving efficient compression for ignition, its structural integrity is paramount. Defects within the capsule could become significant sources of instabilities that impede the desired uniform and symmetric implosion process, as discussed in prior research including [Hurricane et al., 2023, Casey et al., 2015, Schmitt et al., 2013]. To ensure an optimal implosion, it is thus critical that the capsule's surface is uniformly smooth and devoid of any microscopic irregularities such as pits or foreign particles.

The existing methodology for fabricating the capsule involves embedding a layer of tungsten-doped HDC between two undoped layers, utilizing a plasma-assisted chemical vapor deposition process on a spherical, ultra-smooth silicon mandrel, which is subsequently removed. This manufacturing process ensures the inner surface of the capsule meets the requisite surface quality specifications, albeit leaving the outer HDC surface significantly rough. Consequently, this necessitates a subsequent precision polishing process to refine the outer surface to meet ignition quality standards – a detailed approach reported by [Schmitt et al., 2013, Biener et al., 2009]. This intricate polishing procedure not only aims at achieving the required surface smoothness but also corrects the dimensions to achieve precise specifications for ICF experiments.

Leveraging hydrodynamic simulations enriched with empirical data from NIF experiments provides a deeper insight into how initial defects might amplify into significant instabilities during the implosion process and impact the fusion yield. Such simulations, depicted in the works of [Clark et al., 2019, 2013], are instrumental in establishing specifications regarding the permissible size and number of defects, aiming at the minimization of pits to boost overall performance.

As shown in Figure 1, the established protocol for crafting ultrasmooth HDC surfaces comprises an initial lapping process followed by subsequent ultra-precision polishing stages. This regimen is akin to processes used in the preparation of gem-quality materials and aims at not just reducing the initial surface roughness but also achieving accurate diameter tolerances and infusing the final surface with a spectacular finish – the average roughness (Sa) being on the order of a few nanometers, as has been previously discussed.

Despite the ancient heritage and extensive application of mechanical polishing, our comprehensive understanding of the underlying phenomena remains unsettled, rendering the outcomes of ultrafine polishing significantly unpredictable. This unpredictability underlines the importance of implementing real-time monitoring techniques to identify potential defect-inducing anomalies at their nascent stages and to determine the optimal termination point for the polishing process. Advancements in micro-electro-mechanical systems (MEMs)-based sensor technologies and artificial intelligence now facilitate high resolution, real-time monitoring of the polishing process. This technological leap enables the precise identification of anomalies and the determination of process endpoints, as discussed in this paper.

Accordingly, we introduce herein a novel dataset, comprising raw vibration signals emanating from the machinery engaged in the surface polishing of the spheres, collected using an Accelerometer, and surface roughness measurements during the various polishing stages. This dataset is instrumental in automating the assessment of surface roughness. Moreover, it holds the potential to support a broad range of research efforts in material science, physics, and computer science—fields that are essential to the multi-billion-dollar nuclear fusion industry. This dataset is posited as an invaluable resource for researchers dedicated to refining surface polishing techniques and understanding the impact of surface characteristics on HDC capsule performance. In concert with this data, we delineate our evaluative approach employing advanced machine learning techniques to dissect different aspects of this methodical approach, thereby providing a comprehensive overview of our concerted efforts to refine the polishing process and ensure the production of capsules of the highest structural integrity. An automated method to predict surface roughness directly from vibration signals can eliminate the need for human intervention to stop polishing and the time-consuming process of manual surface roughness measurements. Along with this dataset we provide some experiment baselines with neural network models for prediction of surface roughness from vibration data. To generalize the prediction models to different polishing conditions, we also apply domain adaptation techniques to improve prediction accuracy for conditions unseen by the trained model.

The paper is organized as follows: section 2 presents the related work to our dataset and the baseline methods that we used, section 3 explains the dataset and its construction process, section 4 contains the proposed baseline experiments on the dataset, section 5 highlights some limitations of the dataset, and last but not least section 6 concludes the paper and summarizes the important takeaways.

## 2    Related Work

Investigations into the use of vibration data for applications such as predicting surface roughness have gained traction across various engineering fields. However, the specific application of this data within nuclear fusion and precision manufacturing remains largely unexplored.

### 2.1    Historical use of vibration data in process monitoring

The foundational work on the application of vibration data in manufacturing started with Hetherington et al. [1999], who examined its use in monitoring surface conditions of dielectric wafers. While promising, this approach faced limitations in nanoscale applications, where precise cessation thresholds are crucial. Subsequent research has advanced towards more sophisticated models that integrate in-process conditions with surface quality assessments: Bukkapatnam et al. [2008] and Kong et al. [2011] explored regression and Bayesian models to correlate process parameters with vibration characteristics. Advanced signal processing methods like wavelet packet decomposition and polynomial regression were utilized by Garcia Plaza et al. [2018] and Plaza and López [2018] to enhance the understanding of these correlations.

### 2.2    Recent advancements in vibration data and use of ML in manufacturing processes

By utilizing vibration data Jin et al. [2023b] implemented hypothesis testing on bearing area curves to inform decisions in surface finishing of non-flat geometries. Hanchate et al. [2023] and Botcha et al. [2018] demonstrated the application of vibration data to predict surface roughness and monitor dynamics in smart grinding and cylindrical plunge grinding processes, respectively. Real-time process monitoring using vibration data alongside video sensors was notably enhanced by Galla et al. [2024], who applied machine learning and Explainable-AI to distinguish between normal and anomalous interactions during shell polishing. Jin et al. [2023a] developed a statistical model that effectively estimates surface roughness and improves process understanding but falls short in making predictions during the process, illustrating the challenge of implementing real-time predictive capabilities in these contexts. Alexos et al. [2023] also utilized Machine Learning techniques for surface roughness prediction of polishing spheres. While all these studies have utilized vibration signals and machine learning in the context of shell polishing and other precision manufacturing processes, there remains a pressing need for benchmarking and standardization of data to address diverse problems in this domain.

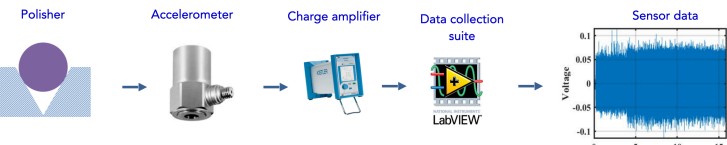

Figure 2: Overview of data collection process

## 2.3 Contribution and impact of the presented dataset

Our dataset paper introduces a uniquely curated dataset for polishing diamond spheres in nuclear fusion technology, accompanied by advanced machine learning models that establish new standards in predictive modeling and real-time anomaly detection. By providing this dataset, we enable the development and validation of machine learning models that achieve high accuracy in surface roughness prediction and facilitate domain-specific analyses. This initiative not only fills a critical gap in the available data resources but also sets a benchmark for future innovations in the field, promoting a deeper understanding and enhanced control over precision manufacturing processes.

# 3 Dataset Description and Construction

In this section, we describe the data and the process with which we gathered them and constructed them. Figure 2 provides the complete overview.

## 3.1 HDC deposition and polishing setup

High-density carbon (HDC) was deposited in a microwave plasma-enhanced CVD (MPECVD) reactor. After coating, capsules were polished in a proprietary-design v-groove polisher with a diamond grinding disk as described in previous work Biener et al. [2009]. During the polishing process, the instrument was stopped periodically to inspect the surface roughness of the HDC capsule. An accelerometer (Kistler K-Shear 8702B500) was attached to the polishing motor body using dental epoxy in order to collect vibrational data from the process. The accelerometer data was collected via a computer-connected amplifier and controlled using custom software written in Labview. Data was generated as a voltage vs. time signal.

### 3.1.1 Baseline polishing dataset collection

Two batches were polished in and measured using standard conditions to complete the baseline dataset. The first batch underwent polishing in 24 hour increments over a total of four stages. At each stage, accelerometer sensor data was collected at a 10kHz sampling rate. Data was collected over the entire run, generating 24 hour time-series datasets. Additionally, a second batch was polished with extra focus on the early-in-time changes to the surface morphology. The polishing process was interrupted every 0.1 hours and shells were removed from the polisher, cleaned by sonication in solvent, and surface roughness data was collected. A total of 18 0.1 hour accelerometer/surface roughness data pairs were collected before the batch was subsequently polished for an additional 72 hours in 12 hour increments.

### 3.1.2 Test dataset collection

A total of three batches were polished to generate the test dataset. Each batch was polished using the same coating and polishing conditions as the baseline batches and vibrational data was collected. The three batches were each polished for 96-100 hours in 24 or 25 hour increments. For one of the batches, surface roughness data was collected every 30 minutes over 1.5 hours and paired with a vibrational spectra dataset to collect early-in-time changes.

### 3.1.3 Manual surface roughness measurements

Surface roughness was measured optically using a scanning laser confocal microscope (Keyence VK-X3100) with a 100x objective lens and a 2mm working distance. A single image measuring approximately 144μm x 108μm was taken on 3-6 capsules for each polishing stage. The surface roughness measurement was made using the Keyence Multi-file Analyzer software. Two filters, a short-pass and long-pass frequency filter, were used to eliminate features smaller than 500nm

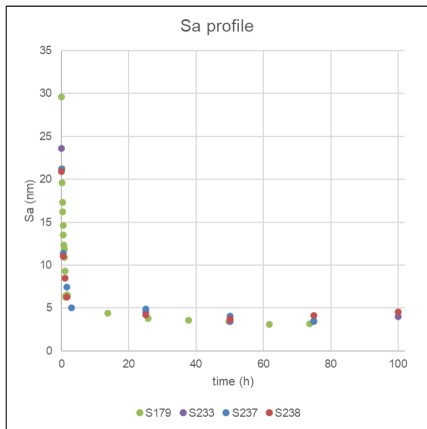

Figure 3: The figure presents the progress of the surface roughness during the polishing process for four different batches. We observe a sharp linear decline in the first hour, and a gradual log decline for the rest of the process. Our motivation for labelling the 6-minute samples lies in this observation.

(approximately 3 pixels) and larger than 25μm (approximately 25% of the short axis of the image). The short-pass filter was used to reduce noise below the resolution limit of the microscope while the long-pass filter was used to eliminate lens artifacts which dominate the surface roughness measurements at very low $S_a$ values. Additionally, a spherical surface correction (F-operation) was used to flatten the data and prevent the curvature of the HDC capsule from affecting the surface roughness data.

## 4 Experiments

To guide practitioners on how to use our dataset, we provide an example ML workflow and some baselines for the regression task described in the previous section. The baselines we provide in this section are based on the extracted statistical features from the raw vibration signals.

### 4.1 Experiment outline

Our dataset presents a regression problem with mapping from raw vibration signals to a change of surface roughness value (deltaSa).

As noted, vibration data, denoted by $x_t$ for $t \in \{1, \ldots, T\}$, is collected at each stage of the experiment using an accelerometer. The data is captured continuously throughout the experiment with $T$ representing the total duration, and is collected at a high resolution with a sampling rate of 10 kHz, corresponding to a sampling interval of 0.0001 seconds.

Surface roughness measurements are captured at the start and end of each polishing stage, which typically are 12-hr or 24-hr long. However, in order to create sufficient data to train an ML model, we segment each vibration data into 6-minute samples and then for each sample, the surface roughness measurements are interpolated to determine the corresponding $Sa$ and $\Delta Sa$ values. Based on the real polishing trends seen in Figure 3 for several polishing batches, our interpolation utilizes a model that integrates both linear and logarithmic changes to reflect the observed data trends: we approximate a linear change in surface roughness during the first hour of the polishing run, followed by a logarithmic change for the remainder of the process. Additionally, we see that the major changes happening in the first hour in Figure 3; on an average we saw a 64% decline in Sa in the first hour for the depicted runs. We assume the same 64% decline during our interpolation.

Over a 24-hour polishing stage, this results in a total of 240 vibration samples, each paired with their corresponding $\Delta Sa$ values determined using our above interpolation method. From each 6-minute vibration sample, 11 statistical features are extracted to characterize the data: kurtosis, skewness, variance, mean, peak acceleration, RMS acceleration, crest factor, shape factor, entropy, impulse factor, and margin factor. These features are depicted in table 1.

For domain detection, we use a mix of polishing batches, code named as S173, S179, S211, S233 and S238, as our train and test data. We note that each polishing batch consists of multiple polishing

Table 1: Statistical features for vibration data and their formulae

| Feature | Description | Formula |
|---|---|---|
| **Kurtosis** | Measure of the "tailedness" of the probability distribution | $\frac{n\sum(x_i-\bar{x})^4}{(\sum(x_i-\bar{x})^2)^2}$ |
| **Skewness** | Measure of the asymmetry of the probability distribution | $\frac{n\sum(x_i-\bar{x})^3}{(n-1)(n-2)\sigma^3}$ |
| **Variance** | Measure of the dispersion of a set of values | $\frac{1}{n}\sum(x_i-\bar{x})^2$ |
| **Mean** | Average of the values | $\frac{1}{n}\sum x_i$ |
| **Peak Acceleration** | Maximum absolute value in the acceleration signal | $\max|x_i|$ |
| **RMS Acceleration** | Root mean square of the acceleration signal | $\sqrt{\frac{1}{n}\sum x_i^2}$ |
| **Crest Factor** | Ratio of the peak value to the RMS value | $\frac{\text{Peak}}{\text{RMS}}$ |
| **Shape Factor** | Ratio of the RMS value to the mean absolute value | $\frac{\text{RMS}}{\frac{1}{n}\sum|x_i|}$ |
| **Entropy** | Measure of the randomness in the signal | $-\sum p_i\log(p_i)$ |
| **Impulse Factor** | Ratio of the peak value to the mean absolute value | $\frac{\text{Peak}}{\frac{1}{n}\sum|x_i|}$ |
| **Margin Factor** | Ratio of the peak value to the square of the mean value | $\frac{\text{Peak}}{\left(\frac{1}{n}\sum x_i\right)^2}$ |

runs as mentioned in section 3. We formulated this as a multi-class classification problem, utilizing 11 statistical features extracted from each 6-minute vibration data sample to classify the domain. .

For our regression experiments, both with a neural network and the domain adaptation method applied to the same network, we used polishing batches S173, S179, and S211 as training/source data, and S233 and S238 as testing/target data. In the domain adaptation approach, we test and adapt separately for each of the testing data. For polishing batches S173, S211, and S233, we used only the first polishing run of approximately 25 hours where only the start and end Sa values are known. We divided the 25 hours into 6-minute samples and determined intermediate Sa values by assuming a linear change with 64% decline in Sa for the first hour and a logarithmic change for the remaining hours as described in Section 4.1. For S179, we used 18 6-min samples corresponding to the first 108 mins of polishing, followed by a 12-hour polishing run. For the first 18 samples, we got the Sa values/6-min from actual manual measurements but for the next 12-hr run we only had start and end Sa values known, which we then divided it into 6-minute samples and assumed a logarithmic change. For S238, we used four polishing runs: first three runs of 30 minutes each, followed by one run of 23.5 hours. We assumed a linear decline in the first two 30-min runs and a logarithmic decline for the remaining runs.

## 4.2 Domain detection

We also investigate the use of vibration data collected from different polishing batch runs to classify them in separate domains based on the polishing stage. Each domain represents a distinct phase in the polishing process, with unique properties that share similarities in analysis (e.g., the type of polishing plate used during a polishing batch).

Our primary challenge is to accurately classify new polishing data into its corresponding domain. This classification is crucial because each domain requires tailored analysis methods. By classifying the domain of the data using 11 extracted statistical features, we can design and implement appropriate analysis techniques. This approach enhances our understanding of the specific characteristics of each polishing batch, ultimately leading to more precise and effective analysis.

For this domain detection task, we utilized standard classification algorithms such as logistic regression, support vector machines, decision trees, random forest, XGBoost and gradient boosting algorithms. We select the algorithm that gives the best classification accuracy. The key motivation here is that the dataset provided can be formulated and utilized as a multi-class classification problem.

## 4.3 Baseline regression methods

Next, we proposed two baseline methods, both based on neural networks for regression and both utilize the same neural network. For these two approaches we use a multi-layer perceptron (MLP) neural network with 2 hidden layers with 100 neurons each and the output layer has one node since

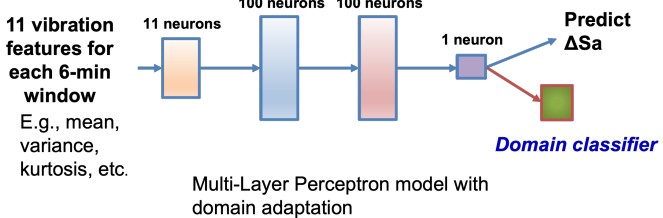

Figure 4: Our MLP-based regression model that predicts change in surface roughness using the vibratio features. The model is also augmented to perform domain adaptation by adding a domain classifier layer that helps to extract features common to both the labeled source polishing domain and the unlabeled target polishing domain.

the problem we are tackling is a regression problem. The first approach directly uses this neural network for regression to predict the change in surface roughness.

The second approach is a domain adaptation technique that tries to consider the domain issue of the data that we described in section 4.2. Briefly, the dataset consists of various polishing runs which can belong to different domains based on the polishing conditions that were used (such as different polishing plates). This applies to both the vibration data as well as the target surface roughness of the polishing runs, which makes the problem we are trying to solve even more challenging. For the domain adaptation approach we utilized the Unsupervised Domain Adaptation by Backpropagation method proposed by Ganin and Lempitsky [2015]. As shown in Figure 4, a neural network model is trained on the labeled data from the source domain (S173, S179, and S211) and is adapted to an unseen unlabeled data from the target domain (S233 or S238). The idea is to augment the neural network with a domain classifier (i.e., a simple gradient reversal layer) that allows the model to extract features common to the two domains. This architecture can be trained by using standard backpropagation, and it is a perfect candidate for our dataset where we utilize unlabeled test data, where both the data and the prediction targets belong to different domains.

We assume the following setup where we have input samples $x \in X$ and regression outputs $y$ coming from a space $Y$. We assume that there exist two distributions $S(x,y)$ and $T(x,y)$ on $X \bigotimes Y$ which refer to source distribution and target distribution (or source and target domains) respectively. Both distributions are assumed to be unknown and different (a domain shift due to different polishing conditions). The goal is to predict the regression outputs $y$ of the target domain $T(x,y)$.

The specific domain adaptation method that we utilized here trains a model with a joint loss function that consists of the regression loss of the labeled source (or training) data, and the binary domain classification of the two domains, source and target. We do not utilize the surface roughness of the target domain for the regression loss because they will not be available to the model in real time during polishing at test time (hence we use an unsupervised adaptation approach). For binary classification loss, we set labels of 0 for source domain data and labels of 1 for target domain data. We define the full loss as:

$$E = L_y + L_d \tag{1}$$

where $L_y$ is the regression loss of the source domain and $L_d$ is the loss of binary domain classification. Together, these losses constitute the total loss that is used in backpropagation during training for domain adaptation.

## 4.4 Results and discussion

**Domain Detection results:** For domain detection tasks, we conducted a comparative study of various learning algorithms as mentioned in 4.2 with only using the vibration features (11 statistical features). The results are presented in Table 2. Our analysis reveals that the gradient boosting algorithm outperformed the others, achieving 100% accuracy in classifying the different domains: each of the polishing batches (S179, S173, S211, S233, and S238) belonged to different domains. This result justifies the need for domain adaptation where testing domains (S233 and S238) are different from the remaining training domain as needed for accurate Sa predictions.

Table 2: Domain detection: Performance comparison of domain classification algorithms

| Sl.No | Algorithm | Macro Precision | Macro Recall | Macro F-1 score | Weighted Precision | Weighted Recall | Weighted F-1 score | Accuracy |
|---|---|---|---|---|---|---|---|---|
| 1 | **Logistic Regression** | 0.26 | 0.4 | 0.3 | 0.3 | 0.49 | 0.35 | 0.49 |
| 2 | **SVM** | 0.46 | 0.6 | 0.5 | 0.31 | 0.48 | 0.36 | 0.48 |
| 3 | **Decision Tree** | 0.93 | 0.99 | 0.95 | 0.99 | 0.99 | 0.99 | 0.99 |
| 4 | **Random Forest** | 0.9 | 0.99 | 0.93 | 1 | 0.99 | 0.99 | 0.99 |
| 5 | **Gradient Bosst** | **1** | **1** | **1** | **1** | **1** | **1** | **1** |
| 6 | **XG Boost** | 0.93 | 0.99 | 0.95 | 0.99 | 0.99 | 0.99 | 0.99 |

**deltaSa prediction results:** As mentioned in earlier sections we conduct a regression prediction based on the 11 statistical features that we extract from the polishing vibration data. We utilize both a neural network (MLP) and a domain adaptation technique on the same neural network. The MLP with 2 hidden layers (100 neurons each) is first trained on the training data (S179, S173, and S211) with Mean Absolute Error (MAE) as the loss function, and tested on S233 and S238 data. The same MLP is then adapted using a combined loss function of MAE and the domain loss as shown in Equation 1. The predicted vs. ground truth Sa results are depicted in fig. 5. We note that the MLP without adaptation achieved a MAE of 0.11nm for the S233 and 0.205nm for the S238. Domain Adaptation, on the other hand, outperformed the plain Neural Network by a big margin achieving 0.0674nm for the S233 and 0.0634nm for the S238.

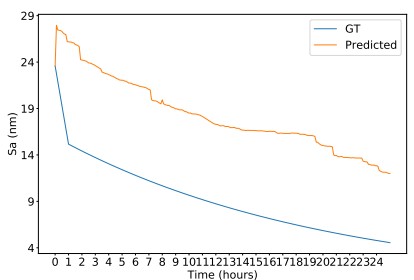

(a) S233 data prediction with the MLP

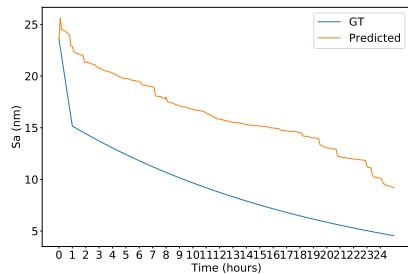

(b) S233 data prediction with Domain Adaptation on the same MLP

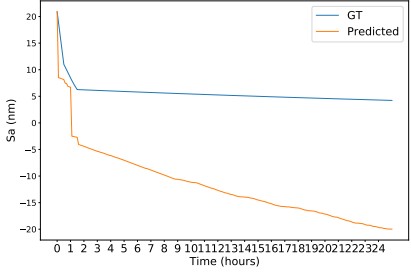

(c) S238 data prediction with the MLP

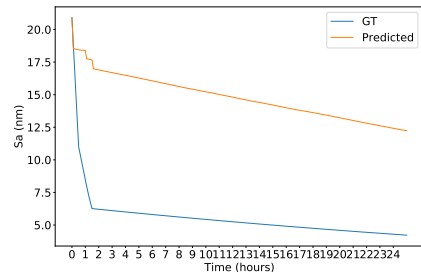

(d) S238 data prediction with Domain Adaptation on the same MLP

Figure 5: Experiments with test data (S233 and S238) with the MLP and the domain adaptation on the same MLP. We observe that domain adaptation performs better than the network without adaptation for both datasets. The adaptation of the trained MLP was performed per run for each of the two test data.

We performed another study where we first adapt the trained MLP on S233 using Domain Adaptation, and then use the newly adapted model for further Domain Adaptation on S238. We observed that the performance on the S238 was slightly better at 0.0618nm. In general Domain Adaptation leads to a lower prediction error just by augmenting the same Neural Network with some extra layers to perform the domain classification task. As we mentioned earlier, the reason that we used Domain Adaptation is because the training polishing condition can be very different from the testing data.

Our experiments above demonstrate how our dataset can be effectively used in predicting deltaSa for different batches of polishing runs, providing substantial insights into the model's ability to generalize across varying conditions.

# 5    Limitations

**Data Augmentation.**    A limitation of the dataset is the methodology we followed to produce more data points. As the Sa values need to be manually measured which is a very time consuming task, it is hard to get a lot of Sa data for robust ML training. Therefore, we decided to split the runs in smaller 6-minute samples and assume a linear and log decline in Sa as described in section 4.1. Although not the most accurate method, our assumption is based on the empirical data corresponding to several polishing batches as shown in Figure 3.

**Extracted Features.**    Another limitation lies in the statistical features that we decided to extract from the raw vibration data. As we mentioned in section 4.1, the 11 statistical features that we decided to utilize might not be very representative of the raw vibration data. Analyzing the raw vibration data is difficult on its own due to its size; $\approx 24$ hours with a sampling rate of 10kHz, which leads to massive files for each polishing run. We therefore decided to extract features in the time domain with statistical features. However, since we provide all of the raw vibration data, the users can utilize other statistical features for modeling as they see fit.

**Data Collection.**    The collection of the raw vibration signal data is a very expensive process and this is the main reason of the limited amount of data that we provide in this first version of this dataset. To overcome this temporary limitation we tried augmenting the data as mentioned earlier in this section by splitting each polishing run on 6-minute samples. From the baseline experiments, we observe that the data we have provided in this dataset along with the data splitting is enough for the models to achieve a good performance. We will keep updating our dataset repository with more data as we generate them.

# 6    Conclusions

In this paper, we introduce a novel dataset focused on the surface roughness of Inertial Confinement Fusion (ICF) capsule targets, monitored through an accelerometer during the polishing process. The pivotal insight derived from this work is the potential for vibration signals to serve as real-time proxies for assessing the surface quality of these capsule targets, significantly enhancing efficiency and resource allocation in their preparation for nuclear fusion applications. Polishing of the ICF capsules is a meticulous, multi-stage process extending over several days, demanding substantial resources. Ensuring the process progresses toward optimal outcomes is critical for preventing shell cracking and achieving precise endpoint detection, thereby necessitating the implementation of an optimal stopping criterion. The specific aspect of the polishing process addressed by this dataset is the quantification of shell surface roughness—a procedure traditionally reliant on time-intensive manual assessments. By enabling the prediction of surface roughness measurements from vibration data, the dataset we present holds the promise of streamlining this aspect of capsule preparation, thereby contributing to the advancement of nuclear fusion technology through improved efficiency and resource utilization.

In conjunction with the dataset, we propose methodologies for data utilization, focusing on the condensation of the dataset into a subset of extracted, pertinent features. Accompanying the dataset, we furnish baseline evaluations employing two distinct approaches: a simple neural network model and a domain adaptation technique on the same network. The rationale behind the incorporation of the latter method stems from the observed phenomenon of domain shifts affecting both the source and target data within our dataset. Our hypothesis regarding the advantage of domain adaptation in mitigating the impact of these domain shifts was validated, as evidenced by the superior performance of the domain adaptation method when compared to the conventional neural network approach. This outcome underscores the efficacy of domain adaptation strategies in enhancing the predictability and applicability of the dataset under conditions of domain variability in polishing settings across different batches.

# 7   Acknowledgements

This work was performed under the auspices of the U.S. Department of Energy by LLNL under contract DE-AC52 -07NA27344 and was supported by the LLNL laboratory-directed research and development (LDRD) program under project 23-ERD-014. We would also like to thank Dan Clark from Lawrence Livermore National Lab (LLNL) for his inputs and for generating Figure 3 and Juergen Biener (LLNL) and Christoph Wild from Diamond Materials GmBH for many helpful discussions.

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
