# OpenReview forum: "Nuclear Fusion Diamond Polishing Dataset"
_NeurIPS.cc/2024/Datasets_and_Benchmarks_Track — NeurIPS 2024 Track Datasets and Benchmarks Poster_

### Official Review · Reviewer_oCrs · 2024-07-11
**Niche work for the ICF community with limited demonstrations**

**Rating:** 6
**Confidence:** 3
**Correctness:** The contributions and claims seem to …

**Review:**

The overall quality of the work is mediocre. While collecting and releasing a dataset is valuable, the machine learning experiments lack depth and rigor. The data collection process and experimental setup are described in reasonable detail, but the machine learning methodology and results analysis are superficial. Key information is scattered throughout, and the authors frequently introduce new concepts or terminology without adequate explanation. Overall, the significance of this work is limited in its current form. While potentially useful for a small community of researchers working on ICF target fabrication, the broader impact is questionable. As mentioned before, the machine learning results do not demonstrate substantial advances in methodology or performance. More importantly, it fails to convince how this improves over existing human in the loop surface roughness studies.

**Strengths:**

The primary strength of this submission is the introduction of a new dataset in a specialized domain. Inertial confinement fusion is an important area of research, and improving target fabrication processes could have meaningful impact. By releasing this dataset, the authors enable other researchers to explore machine learning approaches for this problem. There is potential for the dataset and problem formulation to encourage further research. From an ethical standpoint, research aimed at improving fusion energy technology has positive societal implications. The work does not appear to raise any significant ethical concerns.

**Additional Feedback:**

None

**Clarity:**

The paper is well written but the technical contents from ICF and the data set design is at times hard to follow especially in the parts where data interpolation is discussed.

**Documentation:**

Yes

**Limitations:**

Please see above sections on opportunities for improvement.

**Opportunities For Improvement:**

*The machine learning experiments are shallow and lack rigorous evaluation. Only basic neural network architectures are explored, with little justification for hyperparameter choices.
*The data interpolation approach used to generate more training samples is questionable and not well validated. Assuming linear and logarithmic changes in surface roughness may introduce significant errors based on eye-balling the curves.
*Sample sizes for some experiments seem quite small, raising questions about statistical significance and generalizability of the results.
* The fact they obtain such a high accuracy with gradient boosting begs the question whether the data set is trivial for the application or not.
* For the regression problem on predicting change in surface roughness, it is not clear what is an acceptable level of error from figure 5 and the discussion on "deltaSa prediction results" paragraph. Especially beacuse the predictions are seem to diverge quite a bit after the initial few hours. The domain adaptation approach shows some promise, but the improvements are modest and not well analyzed. The authors do not explore why adaptation helps in this context. Overall, the results section is particularly weak, with little in-depth analysis of the experimental outcomes.
*The authors claim their dataset is the first of its kind, but do not adequately justify this or explain why existing datasets are insufficient. The machine learning techniques applied are standard methods, with little novelty in their application here.

**Relation To Prior Work:**

Relation to prior work is properly discussed

**Summary And Contributions:**

This paper introduces a dataset for monitoring and predicting surface roughness during the polishing of high-density carbon (HDC) spheres used in inertial confinement fusion experiments. The dataset contains raw vibration signals collected during polishing, along with corresponding sparse surface roughness measurements. The authors propose using this data to train machine learning models that can predict surface roughness changes in real-time, potentially eliminating the need for time-consuming manual measurements. Overall, the authors release a novel dataset combining vibration data and surface roughness measurements from HDC sphere polishing. They perform baseline machine learning experiments for predicting surface roughness from vibration data. They apply domain adaptation techniques to improve prediction accuracy across different polishing conditions. The nature of the work is very niche and its technical contributions are rather limited and very preliminary in nature.

---

> ### Author Rebuttal · Authors · 2024-08-16
>
> We thank the reviewer for their time and constructive feedback that was put in this review. We also thank the reviewer for recognizing the significance of our contribution and the dataset towards the nuclear fusion and the machine learning community. We try to answer all the concerns and questions of the reviewer in a form where Q is the reviewer's question/comment and A is our corresponding answer to that.
>
> **Q:The nature of the work is very niche and its technical contributions are rather limited and very preliminary in nature. The overall quality of the work is mediocre. While collecting and releasing a dataset is valuable, the machine learning experiments lack depth and rigor. The data collection process and experimental setup are described in reasonable detail, but the machine learning methodology and results analysis are superficial. Overall, the significance of this work is limited in its current form. While potentially useful for a small community of researchers working on ICF target fabrication, the broader impact is questionable. As mentioned before, the machine learning results do not demonstrate substantial advances in methodology or performance. More importantly, it fails to convince how this improves over existing human in the loop surface roughness studies.**
>
> A: Nuclear fusion is poised to become a major technology to produce energy while minimizing the impact on climate. Polishing the targets used for the fusion process is a critical task where both quality of polished shells as well as volume (number of shells polished per second) are important. Our dataset is the first to help automate and speed up the surface roughness measurements (i.e., a measure of quality) of the polished shells without human intervention. We strongly believe that this dataset will spawn multiple research efforts in this direction from a broad community in nuclear fusion (material scientists, physicists, and computer scientists) – a multi-billion dollar industry. The use of sensors in combination with ML techniques will allow us to make surface roughness predictions in real time – a fast, accurate, and early prediction can be used to ensure that the polishing is moving in the right direction. As this approach scales, it will not only enhance prediction accuracy but also facilitate anomaly detection, significantly impacting the industry’s efficiency and quality control.
>
> Alternatively, the manual process which involves humans in the loop delays the surface roughness measurements as the polisher needs to be stopped, shells are removed from the polisher, surface roughness is measured manually, after which the polishing is resumed. This process cannot be repeated multiple times during polishing (which lasts for multiple days) as it is high overhead and very labor and time intensive.
>
> The purpose of our ML experiments is to expose this dataset or task to automatic AI methods. More comprehensive and deep experiments using advanced ML techniques are considered as future studies. Our preliminary results are expected to function as an inspiration and kick-off of such research. We only provide an example of how this dataset can be used. Our aim was not to use state-of-the-art models. We understand the importance of evaluating benchmarks. However, we would like to respectfully clarify that the primary contribution of a dataset paper is centered on the dataset itself (as per Neurips guidelines). A more comprehensive benchmark of state-of-the-art algorithms will be considered as our future directions and is beyond the scope of this paper.
>
> **Q:Key information is scattered throughout, and the authors frequently introduce new concepts or terminology without adequate explanation.**
>
> A: Can you give us some examples where we introduce new concepts or terminology without adequate explanation? We are happy to explain this better in the final version of the paper.
>
> **Q:The machine learning experiments are shallow and lack rigorous evaluation. Only basic neural network architectures are explored, with little justification for hyperparameter choices. The data interpolation approach used to generate more training samples is questionable and not well validated. Assuming linear and logarithmic changes in surface roughness may introduce significant errors based on eye-balling the curves. Sample sizes for some experiments seem quite small, raising questions about statistical significance and generalizability of the results.**
>
> A: Our dataset consists of 1000s of samples, where each vibration sample is 6-min long. Please note that we have several polishing runs, each 12-24 hours long with multiple GBs of data. The number of samples depends on the sampling window – we used a 6-min window, but the users are free to choose smaller windows to create a larger dataset and to perform a more fine-grained analysis where the surface roughness prediction is carried out at intervals smaller than 6 mins. The train and test target surface roughness for smaller windows can be interpolated assuming the similar linear-log change as depicted in Fig. 3.
> The assumption of linear-log changes in surface roughness is based on data from four multi-hour runs, where surface roughness was manually measured at intervals of 6 minutes and 30-60 minutes. Figure 3 demonstrates that this assumption holds empirically, showing a fast linear change in the first hour, followed by a slower logarithmic change. This pattern, grounded in the physics of the process, has been validated through multiple experimental runs and is consistent with trends observed in other studies, such as Figure 5 in [1].
>
> [1] Jin, S., Bukkapatnam, S., Michael Hayes, S., & Ding, Y. (2023). Vibration signal-assisted endpoint detection for long-stretch, ultraprecision polishing processes. Journal of Manufacturing Science and Engineering, 145(6), 061007.

---

> > ### Author Rebuttal · Authors · 2024-08-16
> >
> > **Q:The fact they obtain such a high accuracy with gradient boosting begs the question whether the data set is trivial for the application or not.**
> >
> > A: The high accuracy with gradient boosting refers to the classification performance for the domain classification, simply if the algorithm detects that a data sample is “in-domain” or “out-domain”. It does not refer to the regression target (surface roughness) that we are trying to predict. We also highlight that the prediction of the regression targets is a very difficult task due to the nature of the problem, and the final prediction needs to be very precise with a very small error (nano-scale) as it is important to have high quality polished shells so that we can achieve higher fusion energy yield.
> >
> >
> > **Q:For the regression problem on predicting change in surface roughness, it is not clear what is an acceptable level of error from figure 5 and the discussion on "deltaSa prediction results" paragraph. Especially because the predictions seem to diverge quite a bit after the initial few hours. The domain adaptation approach shows some promise, but the improvements are modest and not well analyzed. The authors do not explore why adaptation helps in this context. Overall, the results section is particularly weak, with little in-depth analysis of the experimental outcomes. *The authors claim their dataset is the first of its kind, but do not adequately justify this or explain why existing datasets are insufficient. The machine learning techniques applied are standard methods, with little novelty in their application here.***
> >
> > A: The acceptable level of error in prediction of surface roughness (Sa) is +-5nm. While we achieve a small error in prediction on change in Sa (as mentioned in the 2nd paragraph of Section 4.4), our Sa prediction in Fig 5 (b) and (d) can be further improved, perhaps using more advanced ML techniques. However, please note that the scope of this paper is to provide the dataset and not to create or use advanced ML techniques. Our objective in the experiments was to show how to use this dataset. Moreover, another aspect of correctness is that the predicted Sa follows the correct trend – that is, fast change in the initial stage, followed by slower changes. It can be observed that results presented in Fig 5 (b) and (d) follow this trend. An early correct prediction of the trend in real time can help us make sure that the polishing is headed in the right direction.
> >
> > The adaptation helps in this context as the polishing conditions and settings for the different runs can be different, in which case, domain adaptation allows us to improve the model’s performance for the domain-shifted data. For example, the polisher machine used for S179/S173 is different from the one used for S233 and S238. These differences lead to domain shifts in the vibration features whose effects can be minimized using domain adaptation methods. Please note that while the majority of domain adaptation methods are performed on image data, we are the first to demonstrate its effectiveness for vibration data.
> >
> > While there exists vibration data corresponding to manufacturing processes and machines (Sec 2.2), they are not applicable to ICF target polishing as this process is very different from the others. Our dataset is very specific to the ICF target polishing. The vibration signals generated during the polishing of shells capture characteristics of the materials of the shells being polished (e.g., diamond or other HDC material), number of shells polished, polishing settings such as type of grinding disk, etc. Additionally, other datasets do not include measurements of surface roughness, which are part of our dataset and needed to make precise Sa predictions (critical for this application to ensure highest structural integrity of capsules).

---

> > ### Comment · Reviewer_oCrs · 2024-08-23
> > **I will go through the rebuttal**
> >
> > Dear Authors. My apologies for the late reply. I was travelling due to a personal emergency this week. I will go through your rebuttals and will adjust my score accordingly by Monday.

---

> > ### Comment · Reviewer_oCrs · 2024-08-26
> > **Sufficiently satisfied with author rebuttal**
> >
> > Dear authors,
> >
> > After going through your reviews, I respect the importance put on the dataset open sourcing of the work and have updated my score accordingly. I would still encourage authors to properly contextualize the results in figure 5, thoroughly explain, as mentioned in the rebuttal, the significance of the area of work, the importance of the dataset. Also, the data set description should be more accessible to any non ICF related researcher for broader impact.
> >
> > Thank you

---

> > > ### Author Response · Authors · 2024-08-31
> > >
> > > Dear reviewer thank you very much and we appreciate it. We will incorporate your further feedback in the camera-ready version of the paper.

---

### Official Review · Reviewer_yAxf · 2024-08-23

**Rating:** 7
**Confidence:** 3
**Correctness:** Yes.
**Clarity:** Yes.

**Review:**

Clarity: Clear to me

Originality: good, as the authors conduct their own experiments for data collection

Significance: the topic itself is very important because it has connection to energy technology.

**Strengths:**

Pros:
- The study addresses a critical topic with significant relevance to the energy industry.
- The construction of the dataset involved considerable effort.
- The authors have thoroughly presented comprehensive background information."

**Additional Feedback:**

Q@

**Documentation:**

For data collection yes. For data maintenance it is unclear.

**Limitations:**

Yes

**Opportunities For Improvement:**

- The authors should consider emphasizing whether the test samples differ significantly from the training samples, as traditional learning models may already achieve high performance on the test set under such conditions.
- A related question is the model's generalizability when trained on this dataset. Although it might be challenging or impractical, evaluating the model on corner cases could provide valuable insights.
- Once this dataset is established, what practical applications could it serve? Specifically, could it be used to enhance the polishing process, or is there limited potential for further improvement?

**Relation To Prior Work:**

Yes.

**Summary And Contributions:**

This paper introduces the important task of surface polishing spheres for use in nuclear fusion. While many traditional techniques for making these spheres are available and quite mature, machine learning algorithms can be used to predict smoothness for better quality control, potentially saving development time. The authors conducted real-world experiments to collect data, perform data splitting, and benchmark the performance of several traditional algorithms.

---

> ### Author Rebuttal · Authors · 2024-08-27
>
> We thank the reviewer for their time and constructive feedback that was put in this review. We also thank the reviewer for recognizing the significance of our contribution and the dataset towards the nuclear fusion and the machine learning community. We try to answer all the concerns and questions of the reviewer.
>
> **Q: The authors should consider emphasizing whether the test samples differ significantly from the training samples, as traditional learning models may already achieve high performance on the test set under such conditions.**
>
> A: The test and train samples differ significantly and we consider that they are part of different domains since they belong to different polishing runs (and use different polishers with varying settings such as polishing plate). For example, the polisher machine used for S179/S173 is different from the one used for S233 and S238. Due to these challenging conditions, we have deployed a domain adaptation technique for our data. Please also note that while the majority of domain adaptation methods are performed on image data, we are the first to demonstrate its effectiveness for vibration data.
>
> **Q: A related question is the model's generalizability when trained on this dataset. Although it might be challenging or impractical, evaluating the model on corner cases could provide valuable insights.**
>
> A: As we have mentioned each polishing run produces different data (both input and target for each run) so the good performance of domain adaptation proves its generalizability to various data. The testing (and adapting) of our model on a domain different from the training one covers several corner cases and allows us to perform robust testing.
>
> **Q: Once this dataset is established, what practical applications could it serve? Specifically, could it be used to enhance the polishing process, or is there limited potential for further improvement?**
>
> A: The main application that we are targeting is the enhancement of the polishing process in nuclear fusion, and more specifically to automate and speed up the surface roughness measurements (i.e., a measure of quality) of the polished shells without human intervention. We strongly believe that this dataset will spawn multiple research efforts in this direction from a broad community in nuclear fusion (material scientists, physicists, and computer scientists) – a multi-billion dollar industry. The use of sensors in combination with ML techniques will allow us to make surface roughness predictions in real time – a fast, accurate, and early prediction can be used to ensure that the polishing is moving in the right direction. The major impact lies in when this process becomes scalable in the future. As this approach scales, it will not only enhance prediction accuracy but also facilitate anomaly detection, significantly impacting the industry’s efficiency and quality control. High quality and smooth target shells are needed to achieve net fusion yield (please see Sec 1 of the paper for more details on this motivation).
>
> Alternatively, the current manual process which involves humans in the loop delays the surface roughness measurements as the polisher needs to be stopped, shells are removed from the polisher, surface roughness is measured manually, after which the polishing is resumed. This manual process cannot be repeated multiple times during polishing (which lasts for multiple days) to perform close monitoring of polishing as it is high overhead and very labor and time intensive.

---

### Official Review · Reviewer_Gqfh · 2024-08-24
**Review for Nuclear Fusion Diamond Polishing Dataset**

**Rating:** 5
**Confidence:** 3

**Review:**

The authors employ machine learning techniques, including neural networks and domain adaptation methods, to predict changes in surface roughness from raw vibration data captured during the polishing process. The dataset includes raw vibration signals, extracted statistical features, and associated surface roughness metrics, offering a novel resource for the research community. The construnction for the dataset is clear, but the MLP model need more clarification. This dataset could potentially streamline the ICF capsule polishing process.

**Strengths:**

- Novel Application: The paper explores a new application of machine learning in the field of fusion energy
- New Dataset: The release of a novel dataset specifically designed for polishing optimization is a valuable contribution to the research community.
-Model Application: The used of Domain Adaptation outperform the naive MLP

**Additional Feedback:**

What would be your plan to release additional data?

**Clarity:**

The paper is clear on the dataset construction. The model might need more clarification, see in Opportunities For Improvement.

**Correctness:**

The authors construcuted the dataset in a sound way. They have a detail explanation on the data collection and the augmentation.

**Documentation:**

Yes

**Ethics:**

No, there are no ethics concerns

**Limitations:**

The authors addressed the limitations, including limited data scale, suboptimal data augmentation, and potential improvement for feature extraction. They stated they would continue update more data to resolves the data scale issue.

**Opportunities For Improvement:**

- Quality and Clarity: The proposed model includes some aspects that are confusing and could benefit from further clarification. The model is composed of two main components: an MLP (Multilayer Perceptron) and a domain classifier. In the loss function (Equation 1), $L_y$ ​
  is defined as the regression loss for the source domain, while $L_d$ is the classification loss for distinguishing between the source and target domains.  This implies that the training data must include vibration features from both the source and target domains. There are a few areas needing more explanation:

1) For the **target domain** training data, how is $L_y$ defined when training? Is it set to 0, or is there a specific formulation used?

2) Given that the model requires vibration features from the target domain, does this mean that every time the algorithm is applied to a new domain, it must be retrained or fine-tuned online? This could limit the model's practical applicability.


- Clarity and Significance: The scale in Figure 5 appears to be misaligned with the descriptions in the text. For example, in Section 4.4, the authors state that the Mean Absolute Error (MAE) is 0.11 nm for S233 using the MLP, but the graphic seems to indicate a value closer to 11 mm. It's important to ensure that the scales in figures are consistent with the textual descriptions to avoid confusion.

- (Minor Concerns) The features seems unlabled in the dataset (https://mlphysics.ics.uci.edu/data/2024_nuclear_fusion_diamonds/extracted_statistical_features/features_S179_pol1_12hr.txt). It would be helpful to provide a detailed explanation of what each column represents or direct readers to where this information can be found. This additional clarity would enhance the usability of the dataset for other researchers.

**Relation To Prior Work:**

The authors clearly stated this is the first dataset to the public that consist raw vibrationsignals and corresponding polishing surface roughness.

**Summary And Contributions:**

The paper introduces a novel dataset designed to optimize the polishing process of inertial confinement fusion (ICF) capsules using machine learning. The primary contributions of the paper are:
- Novel Dataset Release: The paper presents a newly developed dataset specifically tailored for the polishing optimization of ICF capsules. This dataset is crucial for advancing research in the precise polishing needed for ICF experiments.
- MLP Algorithm with Domain Adaptation: The authors propose a multilayer perceptron (MLP) algorithm that incorporates domain adaptation techniques to account for different polishing conditions. This adaptation allows the algorithm to generalize better across varying environments and conditions, leading to improved performance in the polishing process.

---

> ### Author Rebuttal · Authors · 2024-08-27
>
> We thank the reviewer for their time and constructive feedback that was put in this review. We also thank the reviewer for recognizing the significance of our contribution and the dataset towards the nuclear fusion and the machine learning community. We try to answer all the concerns and questions of the reviewer.
>
> **Q: Quality and Clarity: The proposed model includes some aspects that are confusing and could benefit from further clarification. The model is composed of two main components: an MLP (Multilayer Perceptron) and a domain classifier. In the loss function (Equation 1), Ly​ is defined as the regression loss for the source domain, while Ld is the classification loss for distinguishing between the source and target domains. This implies that the training data must include vibration features from both the source and target domains. There are a few areas needing more explanation:
> 1)For the target domain training data, how is Ld defined when training? Is it set to 0, or is there a specific formulation used?
> 2)Given that the model requires vibration features from the target domain, does this mean that every time the algorithm is applied to a new domain, it must be retrained or fine-tuned online? This could limit the model's practical applicability.**
>
> A: Regarding 1, for Ld we define with 1s the samples that we consider source domain and with 0s the samples that we consider target domain. The domain classifier of the model is trying to classify then if the data points belong in the source domain or in the target domain. Ld as we mentioned is just a binary cross entropy loss for binary classification (source and target domain).
> Regarding 2,  Only in cases when domain shift is detected between source and target domains. In our case, such a shift was observed as different polishers and polishing conditions were used for different runs.  For example, the polisher machine used for S179/S173 is different from the one used for S233 and S238.
>
> An automated method, such as using an autoencoder, can also be used to perform domain shift detection. This is out of scope for the paper as we primarily focused on releasing the dataset and showing some preliminary experiments to demonstrate how to use it.
>
> **Q: Clarity and Significance: The scale in Figure 5 appears to be misaligned with the descriptions in the text. For example, in Section 4.4, the authors state that the Mean Absolute Error (MAE) is 0.11 nm for S233 using the MLP, but the graphic seems to indicate a value closer to 11 mm. It's important to ensure that the scales in figures are consistent with the textual descriptions to avoid confusion.**
>
> A: The misconception of this observation lies in the fact that in the text of subsection 4.4 we mentioned that the average deltaSa of S233 (i.e., the change in Sa or surface roughness during each 6-min duration) is 0.11 nm while Figure 5 depicts the Sa or actual surface roughness values. We decided to depict the Sa and not the deltaSa in Figure 5 as the former provides more information about the overall polishing process as the time progresses and one can visualize the trend better. Presenting deltaSa looked non-informative and we decided that it would not help the audience better understand our predictions. But based on your comment and to avoid any confusion, we can also add plots on deltaSa ground truth vs. predictions in the final paper.
>
> **Q: (Minor Concerns) The features seems unlabeled in the dataset (https://mlphysics.ics.uci.edu/data/2024_nuclear_fusion_diamonds/extracted_statistical_features/features_S179_pol1_12hr.txt). It would be helpful to provide a detailed explanation of what each column represents or direct readers to where this information can be found. This additional clarity would enhance the usability of the dataset for other researchers.**
>
> A: We will make sure to add these details for the research community. For clarification each column is a statistical feature (mean, kurtosis, etc.) that we extracted from the raw vibration signals. We have mentioned which statistical features we have extracted in Table 1 of the main text.
>
> We hope that we have answered all of the reviewer's concerns and questions. If there is anything else please let us know during the discussion period.

---

> > ### Author Rebuttal · Authors · 2024-08-29
> >
> > **Q: What would be your plan to release additional data?**
> >
> > A: At this stage this is the best effort that can be done towards the data collection. Again as we have described in earlier sections of our manuscript, data collection is a very time-consuming and expensive process. A larger dataset will require a tremendous amount of work, but we are committed to explore that in the future. For now we consider the current dataset as a version 1, and once we can collect more data we can start building a version 2. Other researchers have moved in a similar direction. Regardless we believe that the current version is sufficient in its form, and valuable for the research community.

---

> > > ### Author Rebuttal · Authors · 2024-08-31
> > >
> > > Thank you for taking the time to review our work. If you have any further concerns after reading our response, please don’t hesitate to reach out. We would be glad to address any additional questions or comments you may have. Please also let us know if you have received our responses and whether we have successfully addressed your concerns.
> > > Thank you once again for your valuable insights and feedback.

---

### Decision · Program_Chairs · 2024-09-26

**Decision:**

Accept (Poster)

**Comment:**

The paper presents a novel dataset and applies machine learning techniques to address an important problem in inertial confinement fusion (ICF). The dataset, which includes raw vibration signals and surface roughness measurements from the polishing process of high-density carbon spherical shells, is the first of its kind. The authors propose the use of a multi-layer perceptron (MLP) with domain adaptation to generalize predictions across different polishing conditions. The contributions are highly relevant to a niche yet significant domain, with implications for the nuclear fusion industry.

Pros:
- Novelty: The release of a unique dataset specifically designed for ICF target fabrication is a valuable contribution to the community. The inclusion of both raw and statistical features from vibration signals enhances its potential utility.
- Practical Relevance: Automating surface roughness measurements in the polishing process could significantly improve efficiency, reduce manual labor, and increase the scalability of fusion energy research.
- Rigorous Data Collection: The dataset is carefully curated from real-world experiments, with detailed explanations provided on the polishing runs and conditions.

Cons:
- Clarity Issues: Several reviewers raised concerns about the clarity of the model and certain figures. For instance, the misalignment of scales in Figure 5 led to confusion. The explanation of domain adaptation also requires more detail for better understanding.
- Shallow Analysis: The machine learning experiments, while a valuable demonstration, are preliminary, with limited exploration of more advanced techniques. Further rigor and depth in the experimental setup and hyperparameter choices would enhance the impact.
- Generalizability: Concerns about the model's generalization capabilities under different polishing conditions were noted. The authors addressed these in their rebuttal, but deeper exploration of model robustness and domain adaptation improvements would strengthen the submission.

Given the novelty of the dataset and its potential impact, the paper is worth accepting despite the aforementioned reservations. The authors' response satisfactorily addresses most concerns, and further refinements in the final version will improve the clarity and accessibility of the work. The dataset's release will likely inspire future research in the field.